# Frontline Healthcare Workers’ Reluctance to Access Psychological Support and Wellness Resources During COVID-19

**DOI:** 10.3390/healthcare13222887

**Published:** 2025-11-13

**Authors:** Kevin P. Young, Diana L. Kolcz, Jennifer Ferrand, David M. O’Sullivan, Kenneth Robinson

**Affiliations:** 1Psychology/Psychiatry, Institute of Living, Hartford Hospital, Hartford, CT 06102, USA; 2Wellness, Medical Affairs, Hartford HealthCare, Hartford, CT 06102, USA; 3Research Program, Hartford HealthCare, Hartford, CT 06102, USA; 4Department of Emergency Medicine, Hartford Hospital, Hartford, CT 06106, USA

**Keywords:** healthcare worker, reluctance, depression, anxiety, burnout, treatment, support, barriers, front-line, emergency room

## Abstract

**Background/Objectives**: We sought to determine the factors associated with Emergency Department (ED) healthcare workers (HCW) reluctance to seek, utilize, or take advantage of psychological support services during the pandemic. **Methods**: A 53-item survey, delivered via REDCap, was completed by ED staff in seven hospitals between 15 July 2020 and 24 August 2020. **Results**: 351 participants (28.7% response rate) completed the survey with 20.1% of respondents endorsing clinically significant psychiatric symptoms and 31.7% of participants endorsing burnout. 75% of those who endorsed significant emotional symptoms did not seek formal psychological support. Most of those (33/44) who did not seek support, despite anxiety and/or depression, reported experiencing practical barriers (access, cost, time, confidentiality) while emotional barriers (not wanting to acknowledge needing help; stigma; embarrassment) were endorsed by 22.7% (10/44). **Conclusions**: These findings offer several opportunities for intervention, including changes to workflow and culture in the ED which may address emotional barriers to self-care and pragmatic system changes that may help address practical barriers.

## 1. Introduction

Pandemics have varied psychological effects on society, including depression, anxiety, panic attacks, somatic symptoms, posttraumatic stress disorder (PTSD), psychosis, and suicidality [1,2]. The current pandemic poses unique stressors among healthcare workers (HCWs), who are persistently exposed to the effects of the COVID-19 outbreak and are experiencing increased psychiatric symptoms as a result [3,4,5,6].

While all HCWs are at risk for the adverse psychological effects of the pandemic, many studies have reported that frontline workers and Emergency Department (ED) staff are at even greater risk than other HCWs. Following the SARS outbreak in 2002–2003, frontline workers reported greater stress and fatigue, higher levels of depression, anxiety, PTSD, and alcohol consumption, and worse sleep [7]. HCWs in China during the COVID-19 pandemic were found to have a higher risk of symptoms of depression, anxiety, insomnia, and distress [3,8]. Some have indicated that ED staff may be particularly vulnerable to these adverse psychological effects [9,10,11], though discrepant results exist [12].

Research has highlighted the importance of social support, communication, training, peer support, effective coping [10,13,14], and access to mental health/substance use care [15] as means of ameliorating burnout and psychological distress in healthcare. It has been reported that creating resilience in the healthcare industry, either during or outside of a crisis, is a shared responsibility between HCWs and their organizations [14,15]. To mitigate the adverse psychological effects of the pandemic on HCWs and to assist with HCWs’ recovery from these emotional and psychosocial effects, our healthcare system (Hartford HealthCare, which includes 7 acute care hospitals, a behavioral health network with three hospitals, a rehabilitation network, a medical group, and senior and at-home services) developed and instituted many of the strategies promoted elsewhere in the literature [7,8,10,14,16,17,18,19], including enhanced communication, preventative and reactive psychological and psychiatric resources and hotlines, an expanded peer support system, and additional wellness programs.

Paradoxically, even though the pandemic had a significant impact in our region, HCWs rarely took advantage of these resources and programs. In a survey of over 8400 HCWs in our system, more than 80% of participants indicated significant psychiatric symptoms, yet fewer than 8% of those workers sought formal psychological or wellness assistance [19]. In fact, very few frontline workers sought help. If the cause of the underutilization of these resources could be understood, then programs could be redesigned to better meet the needs of HCWs. Our study’s aim was to better understand sources of support and coping used by ED HCWs and to identify the largest obstacles to accessing supports, particularly among those who demonstrated signs of clinically meaningful emotional distress.

## 2. Materials and Methods

Design/Setting: Between 15 July 2020 and 24 August 2020, 1223 ED staff received study invitations (28.7% response rate). After consent, participants completed a 53-item REDCap survey (with branching logic), created by paper authors for use in this study, approved by the Hartford HealthCare Institutional Review Board (HHC-2020-0187).

Measurement: The survey, designed for this study by the authors, included questions assessing demographic characteristics, psychiatric history, burnout severity (via a single-item burnout measure), extent of stressors and supports, engagement with services, and formal measures of depression and anxiety such as the Patient Health Questionnaire-9 (PHQ-9) and General Anxiety Disorder-7 (GAD-7) were used. Assessment of stressors and supports, and measurement of engagement with services, was conducted via pre-populated options (with option for “other”) which were author-derived.

Inclusion Criteria/Participation: Providers, nurses and technicians in the EDs were invited to participate in the study. After an introductory email from the study staff, local ED supervisors were asked to send two reminder emails to all eligible employees in their area; emails were approved by IRB. Neither employee supervisory staff nor hospital administration had any knowledge of which HCWs participated or declined to participate in the research and all participants had their responses protected under standard research protocols.

Analysis: All data were analyzed with SPSS v. 26 (IBM; Armonk, NY, USA, 2019), and all results resulting in *p* < 0.05 were deemed statistically significant. Categorical comparisons were evaluated with a chi square test. Continuous data were evaluated for distribution and analyzed with Student’s *t*-test or Mann–Whitney U test for two groups, and analysis of variance or Kruskal–Wallis H test for >2 groups, depending on distribution. Correlations were evaluated with a Spearman rank correlation coefficient. Since the number of responses was not known initially, no a priori power analysis was performed.

## 3. Results

### 3.1. Overall Study Sample

In total, 351 individuals consented to participate in the survey. The majority (72.3%) of participants were female, and 10.9% were Hispanic. A total of 89% of participants identified as white, 3.7% identified as African American, 2.3% as Asian, and 4.7% answered more than one race. A total of 58% were married, 35% were single, 5.5% divorced, and 1.2% separated or widowed.

The mean GAD-7 score (±standard deviation, SD) was 4.71 ± 5.04, with a median (interquartile range) of 3 (0–7), and the mean PHQ-9 score was 4.26 ± 4.70. Overall, 20.1% (59/294) of participants who completed at least one of these measures endorsed psychiatric symptoms in the clinical range (>9 on either measure, consistent with “moderate” symptoms) per standard interpretation procedures for these instruments. In addition, 31.7% endorsed at least one symptom of burnout, identified by choosing a response other than, “I enjoy my work, I have no symptoms of burnout” or “Occasionally I am under stress, and I don’t always have as much energy as I once did, but I don’t feel burned out” (Table 1). This is a standard method of interpreting results from this measure, per measure authors [20].

The survey asked about sources of emotional support via a pre-populated list of options, and more than one answer was permitted. Over the entire group of respondents to the survey, the most common response was family (70.7%), followed by friends (59.8%), and work peers (49.9%) (Table 2). The “other” category was not further analyzed as it only had a single response in the positive direction; this also suggested that the survey’s pre-populated options were relatively exhaustive.

The survey asked the entire sample to reflect on the types of support that they relied on during this period of time (Table 3). Only 13% indicated that they have sought out support during this time; of these, the primary means of support were reading articles (n = 20, 6%), video meetings with a behavioral health specialist (n = 18, 5%), and 10 people (3%) indicated that they used calls to formal support systems or in-person appointments with behavioral health specialists during this time.

In total, 13.3% of all participants reported that they had actively sought out help during this time. Those who answered that they had not actively sought help (86.7%) were asked why they had not sought emotional support. The most common reason given was, “I don’t believe I need it” (44.7%), followed by “I don’t have time” (21.1%) (Figure 1).

### 3.2. Findings for Individuals with Signs of Emotional Distress

As noted above, 59 participants reported clinically significant symptoms of depression and/or anxiety. Of these, 25.4% (15 of 59) sought out formal psychological support (e.g., via virtual, in person, employee assistance program (EAP), group meetings, or use of a call center). Thus, 74.6% (44) with clinically meaningful symptoms did not seek formal support.

A primary goal of this study was to identify why individuals affected by emotional symptoms did not engage with treatment options. To determine why individuals who were experiencing emotional pain failed to seek formal help (44 of 59 respondents with significant symptoms), we divided the concerns into two intuitive groups: practical barriers (access, cost, time, concerns of confidentiality) and emotional barriers (“I might need it but don’t want to believe that I do”, stigma, and/or embarrassment). Three quarters (33 of the 44) with clinically significant symptoms who did not seek formal help endorsed practical barriers, 22.7% (10/44) endorsed emotional barriers, and 18.2% (8/44) endorsed both (Figure 1).

As one would expect, when we compared reasons for not seeking help to the severity of psychiatric symptoms indicated on the PHQ-9 and GAD-7, individuals who believed they did not need help had mean GAD-7 and PHQ-9 scores that were significantly lower (*p* < 0.001 for both) than individuals who answered they thought they did need help.

Those who indicated other reasons for not seeking help (e.g., not enough time, cost, might need help but don’t want to believe it’s needed, confidentiality/impact on job, stigma, unsure how to access help and embarrassment) had higher mean GAD-7 and PHQ-9 scores (*p* < 0.001 for both) compared to those who did not think that specific reason was a factor for them not seeking help (Figure 2).

Table 4 evaluates correlations between anxiety (GAD7) and depression (PHQ9) independently, with each of 12 coping behaviors, across the entire study sample. Subjects had the following response options for each item, related to frequency of utilization: daily, several times a week, weekly, monthly, or never/almost never. Items with positive correlations reflect the finding that less frequent use of the behavior correlates with higher levels of anxiety and depression; items with negative correlations reflect the finding that more frequent use of that coping behavior is correlated with higher levels of depression and/or anxiety (Table 4). In the overall sample, spending time with family (*p* < 0.001), friends (*p* < 0.05 anxiety and *p* < 0.01 depression), and exercising (*p* < 0.01) were all associated with lower levels of psychiatric symptoms. In the overall sample, drinking at home (*p* < 0.01) and working more (*p* < 0.01) were both associated with reports of higher levels of both anxiety and depression.

Finally, we divided the study sample into whether or not the HCW sought support. Regarding anxiety, those who responded “no” to seeking support had similar results to the overall population, though time with friends was no longer significantly correlated with anxiety symptoms. Those who did seek support indicated that their use of the following coping skills was associated with decreased anxiety: time with family (rho = 0.413, *p* = 0.01), time with friends (rho = 0.406, *p* = 0.011), exercise (rho = 0.528, *p* < 0.001), and reading (rho = 0.350, *p* < 0.05).

Similarly, when coping behaviors and depression were evaluated, time with family and friends and exercise were associated with lessened depression symptoms, while working more and drinking were associated with worse symptoms (data above). These same findings were present for individuals who reported not seeking support. For those who did seek support, time with family (rho = 0.411, *p* = 0.010), exercise (rho = 0.461, *p* < 0.01), and sleeping (rho = 0.337, *p* < 0.05) were all associated with fewer symptoms of depression, while frequenting bars/nightlife (rho = −0.396, *p* < 0.05) and working more (rho = −0.454, *p* < 0.01) were both associated with reports of more symptoms of depression.

## 4. Discussion

### 4.1. Anxiety and Depression Among ED Staff Compared to Others

Almost 15 percent of participants reported moderate to severe anxiety (GAD-7 ≥ 10), which is lower than that of our system-wide study (31%) [21], our national data (33%) [5], a survey of the public across the world (31.9%) [22], and is much lower compared to published rates in China (45%) [3] but greater than the prevalence of generalized anxiety in the general population outside of the pandemic (3%) [23].

Almost 16% of participants endorsed moderate to severe depressive symptoms (PHQ-9 ≥ 10). This is much lower than the rate we found with our system-wide study (83%) [21], the 50% in China [3], and the 33.7% in the world population [22]. This rate is consistent with the 14% prevalence of depression reported in our national survey [5].

Overall, 20% of participants endorsed clinically significant symptoms of anxiety and/or depression, as compared to nearly 40% in our national survey, which included a broad swath of healthcare workers and was not limited to just those in the ED [5]. This is elevated from non-pandemic levels of psychiatric distress, though ED workers in our study endorsed fewer symptoms than HCWs in other COVID-19 studies. It may be that they experience less distress, or possibly are less likely to report the distress that they do experience. Given that 62% of respondents who did not seek help indicated that they did not believe they need it, it is possible that some ED HCWs have limited awareness of their needs for support. Alternatively, it is possible that ED HCWs, like others in high-intensity professions (police, military) are trained and socialized to work through extremely high stress situations without attending to their own physical or emotional needs; their threshold or breaking point may be higher.

### 4.2. Burnout

Increased distress, ranging from feelings of stress to frank symptoms of burnout, was endorsed by 88% of our sample. Following the direction of the authors of our single-item burnout measure [20], we dichotomized the responses to being absent (denied symptoms of burnout but may have endorsed stress) or present (endorsed one or more symptoms of burnout). Similarly to the 38.5% of individuals in the validation sample of this tool who endorsed at least one symptom of burnout (n = 5404; providers, nurses, clinical staff, administrative staff), 31.7% of ED staff endorsed burnout in our survey. Although the overall stress and burnout levels are high, these data suggest that during this pandemic, ED workers may experience similar or slightly lower levels of burnout than other HCW colleagues or, alternately, may be less likely to endorse these symptoms.

Burnout is a real threat to HCWs and is even more so during times of pandemics or widespread infectious disease outbreaks [24,25]. While only 12% of ED workers in this study denied feelings of burnout, 8.3% reported severe levels of burnout and the majority reported at least mild to moderate feelings of burnout. This is consistent with a study by the American College of Emergency Physicians (ACEP) in early October 2020, in which 87% of Emergency Physicians reported feeling more stress since the start of COVID-19 and 72% endorsed greater professional burnout [26]. A concerning issue is that while many reported at least some level of burnout, only 10% reported that they sought emotional support from formal programs at work (e.g., EAP, support groups, consultation with organization mental health workers) or from a private therapist or mental health professional. It is possible that these HCWs derive adequate support from non-professional sources, or that they do not consider these formal work-sanctioned programs to be capable of meeting their needs.

### 4.3. Actively Sought Support

Respondents were asked whether they had actively sought psychological or emotional support or help, including not only formal meetings with behavioral health professionals but also wellness readings and webinars. Only 13.3% reported actively seeking psychological or emotional support. Of the support sought, the most frequent form was “reading articles” (6%), followed by a “video appointment with a behavioral health specialist” (5%).

Perhaps for those who believe they need support and actively seek that support, it may be perceived as more acceptable or values-consistent to obtain support through self-directed education (i.e., reading an article) than in a counseling session. Based on participants’ symptoms of anxiety, depression, and burnout, it would seem that a much greater number of HCWs would benefit from formal support. There may be an opportunity to cultivate a shift in organizational culture such that self-care (demonstrated by HCWs’ willingness to access a broad range of support services) is not only accepted but also reinforced [15].

Similarly, while most reported increased stress or burnout, 44.7% reported that they did not seek help because they did not feel that they needed it. In addition, 9.7% (24/248) of those who indicated that they did not seek formal support endorsed not wanting to admit they might need it. During the peak of the surge, these workers may not have been aware of their feelings or symptoms; they may have felt that feeling distress is a normative experience within a busy ED setting, or perhaps they felt an expectation to get through it on their own. Healthcare leaders might play a role in this situation. If these leaders model a “push through” mentality, fail to recognize the suffering of their staff, or do not provide implicit and explicit permission to seek help, then those values may be transmitted to the staff. Healthcare professionals, and physicians in particular, learn to subjugate their own needs in order to prioritize patient care; this message may be reinforced directly and indirectly by leaders, through several processes (e.g., by focusing on patient satisfaction and achieving the Triple Aim) and by the general public in their efforts to recognize our “Healthcare Heroes.”

One possible solution would be to provide leaders and staff with programs (i.e., resilience training) designed to enhance self-awareness. This may help leaders become more aware of their own feelings and behaviors so they are better able to identify and support distressed staff and cultivate a more accepting attitude toward seeking help. A resilience training program may help staff become more aware of their distress and their needs so as to seek help proactively.

Participants endorsed practical barriers to seeking formal support. Scheduling support time into weekly workflow would normalize, destigmatize, and potentially increase use of services. Related strategies which include increased access to exercise equipment, a “battle buddy” program, or efforts to improve leader resilience might better serve these reluctant HCWs while embedding integral values into the workplace culture.

Participants also indicated cost as a barrier. During the height of the pandemic, ED volumes were down and many physician groups and hospitals furloughed workers. Our organization did not furlough workers and at the same time developed a robust Wellness Department and several wellness initiatives and committees. Multiple times weekly during the height of the surge, staff received emails detailing free and confidential resources, and the organization went so far as to establish and advertise a 24 h call center that provided everything from emotional support to childcare referrals and shopping assistance. It is unclear if reported issues of access and cost reflect a genuine lack of knowledge of system offerings versus deliberate avoidance. One hypothesis is that some of the non-utilizers sought to rationalize their need states to avoid support, which may reflect physician “warrior culture”. This is consistent with data from ACEP that suggests that stigma and job security affect utilization of services [27].

Only 13% reported not seeking help due to feeling uncomfortable or concerned, apprehensive about confidentiality or impact on job (5.1%), associated stigma (5.1%), and embarrassment (2.8%). This is in contrast to the ACEP study in which 45% of emergency physicians reported that they were not comfortable seeking mental health treatment [26]. Additionally, 73% of respondents in the ACEP study felt there is a stigma in their workplace associated with seeking mental health treatment and 57% reported that they feel that their job might be in jeopardy if it were discovered that they sought mental health treatment.

Access to mental health treatment was not found to be a barrier to seeking mental healthcare; only 4.6% reported that they were unsure how to access care. This is similar to the findings in the ACEP study, in which 71% rated their ability to access mental healthcare as good or excellent [26].

We investigated the relationship between levels of anxiety or depression and each of the reasons for not seeking care. Mean GAD-7 and PHQ-9 scores were compared for those who indicated that a specific reason was a factor in not seeking care versus those who did not. For those respondents who reported they did not believe they needed help, both their GAD-7 and PHQ-9 scores were significantly lower, suggesting they may have been accurate in thinking that they do not need help. Individuals who endorsed all other reasons for not seeking help had significantly higher GAD-7 and PHQ-9 scores. This may indicate that these workers had more symptoms, and both the symptoms themselves and the reasons for not seeking care were potential barriers; further investigation is certainly warranted.

## 5. Limitations

The study’s response rate was just under 30%, which is less than ideal, though lower response rates are unsurprising when studying ED HCWs mid-pandemic. Despite this, data regarding symptoms of burnout and emotional distress were generally consistent with published research and use of a self-selecting sample is reasonable and appropriate for real-world, pragmatic research. Our response rate was also similar to the response rate for employee “required” surveys put out by leadership, suggesting that we were at the expected response rate for this population, as opposed to a discrepancy due to research content.

## 6. Conclusions

We found the overall prevalence of depression, anxiety, and burnout to be less severe than we expected, based on previously reported data, both prior to and during this and other pandemics. Overall, 13% of respondents sought help and only 25% of those who reported clinically significant symptoms of depression and/or anxiety sought out formal psychological help. The most common factors associated with the reluctance of ED HCWs to seek, utilize, or take advantage of psychological support services during the pandemic were not believing they needed it, lack of time, and cost. Confidentiality/impact on their job, the potential of associated stigma, and embarrassment were all endorsed less frequently; this may reflect social pressure, versus a genuine belief system, perhaps aided by a system that has prioritized wellness. Findings may not easily be replicated in other samples, which would be meaningful for future research to explore.

This study provides valuable insights to drive the development of suitable support services for HCWs during pandemics, and these insights into the reluctance to seek care are also most likely applicable during non-pandemic times. Though the behavioral health resources and wellness programs developed by healthcare organizations are relevant and potentially helpful to many, we have an opportunity to bring these resources more directly to HCWs in a way that increases their perceived value and in a manner that does not require them to spend non-work hours engaging in these programs. Additional collaboration with Behavioral Health experts and further studies are indicated to determine the most effective means by which to enhance HCWs’ self-awareness of the psychological effects of prolonged stress, to design programs that are more readily accepted and utilized, and to increase the perceived value of these services.

## Figures and Tables

**Figure 1 healthcare-13-02887-f001:**
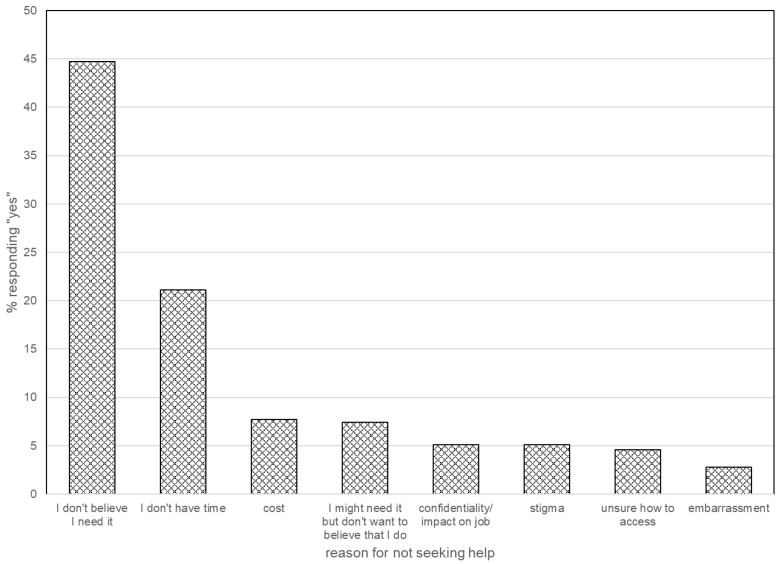
Reasons for not seeking help.

**Figure 2 healthcare-13-02887-f002:**
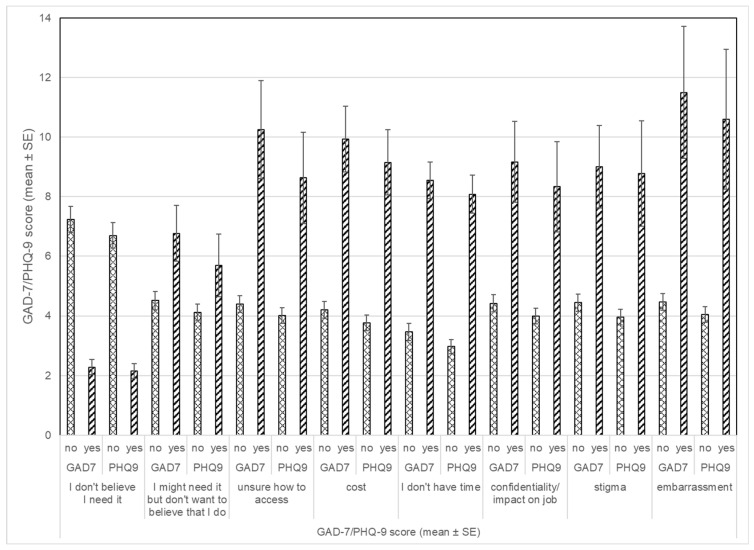
Reason for not seeking care and anxiety and depression scores.

**Table 1 healthcare-13-02887-t001:** Level of perceived burnout.

	No. (%)	
Single-Item Burnout Measure	Yes	No
I enjoy my work. I have no symptoms of burnout.	42 (12%)	309 (88%)
Occasionally I am under stress, and I don’t always have as much energy as I once did, but I don’t feel burned out.	157 (45%)	194 (55%)
I am definitely burning out and have one or more symptoms of burnout, such as physical and emotional exhaustion.	82 (23%)	269 (77%)
The symptoms of burnout that I’m experiencing won’t go away. I think about frustration at work a lot.	21 (6%)	330 (94%)
I feel completely burned out and often wonder if I can go on. I am at the point where I may need some changes or may need to seek some sort of help.	8 (2%)	343 (98%)

**Table 2 healthcare-13-02887-t002:** Sources of emotional support.

	No. (%)	
Type of Support	Yes	No
Family	248 (71%)	103 (29%)
Friends	210 (60%)	141 (40%)
Work leadership	100 (29%)	251 (72%)
Work peers	175 (50%)	176 (50%)
Specialized work programs	11 (3%)	340 (97%)
Religious groups	20 (6%)	331 (94%)
Private therapist/mental health provider	22 (6%)	329 (94%)
Other	4 (1%)	347 (99%)
None	16 (5%)	335 (95%)

**Table 3 healthcare-13-02887-t003:** Type of support actively sought out by participants.

	No. (%)	
	Yes	No
Have you actively sought psychological or emotional support during this time?	38 (13%)	248 (87%)
Type of Support		
Reading articles	20 (6%)	331 (94%)
Phone call to support system	10 (3%)	341 (97%)
Attending webinars targeting wellness/emotional health	8 (2%)	343 (98%)
Video appointments with behavioral health specialist	18 (5%)	333 (95%)
In-person appointments with behavioral health specialist	10 (3%)	341 (97%)
Voluntary participation in groups	3 (0.9%)	348 (99%)

**Table 4 healthcare-13-02887-t004:** Correlations of coping behaviors with anxiety and depression, entire sample.

	GAD-7			PHQ-9		
Coping Behaviors	rho	*p*	n	rho	*p*	n
Time with family	0.215	<0.001	281	0.274	<0.001	283
Time with friends	0.131	0.029	279	0.200	0.001	281
Exercise	0.176	0.003	278	0.273	<0.001	280
TV/Computer	−0.037	0.541	277	0.005	0.938	279
Dining out	−0.011	0.860	276	−0.003	0.956	278
Bars/Nightlife	−0.026	0.667	275	0.037	0.538	277
Drinking at home	−0.229	<0.001	275	−0.187	0.002	277
Reading	0.067	0.265	276	0.070	0.247	278
Sleeping	0.064	0.287	276	0.057	0.342	278
Working more	−0.187	0.002	277	−0.172	0.004	279
Working less	−0.055	0.368	271	−0.030	0.622	273
Traveling/Vacation	0.043	0.479	274	0.118	0.051	276

## Data Availability

The original contributions presented in this study are included in the article. Further inquiries can be directed to the corresponding authors.

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
