# Peer review of "Frontline Healthcare Workers’ Reluctance to Access Psychological Support and Wellness Resources During COVID-19"

_healthcare, 2025, doi:10.3390/healthcare13222887_

Round 1
Reviewer 1 Report
Comments and Suggestions for Authors
AUTHORS sought to determine the factors associated with ED healthcare workers (HCW) reluctance to seek, utilize, or take advantage of psychological support services during the pandemic.
A 53-item survey, delivered via REDCap, was completed by ED staff in 7 hospitals between July 15, 2020 and August 24, 2020.
351 participants (28.7% response rate) completed the survey with 20.1% of respondents endorsing clinically significant psychiatric symptoms and 31.7% of participants endorsing burnout.
Three quarters of those who endorsed significant emotional symptoms did not seek formal psychological support.
Most of those (33/44) who did not seek support, despite anxiety and/or depression, reported experiencing practical barriers (access, cost, time, confidentiality) while emotional barriers (not wanting to acknowledge needing help; stigma; embarrassment) were endorsed by 22.7% (10/44).
AUTHORS concluded that THEIR findings offer several opportunities for intervention, including changes to workflow and culture in the ED which may address emotional barriers to self-care and pragmatic system changes that may help address practical barriers.
The study is interesting and has merit.
I have the following comments for the authors:
1) There are some loose acronyms in the abstract (see ED).
2) The introduction needs improvement: it is very short, and the articles are cited in groups. Furthermore, it doesn't contain a well-defined AIM; it cites a survey as if it were a result, which confuses the reader as to whether it is a previous or current study.
3) The methods are more suited to a conference paper and need to be explained better; each section consists of a few concise, list-style lines.
4) The results are well presented; please include the data labels in the histograms.
5) The discussion is divided into sections but lacks a brief explanation to hold it together. Large sections are not supported by references.
6) For references, use the standard citation []
Reviewer 2 Report
Comments and Suggestions for Authors
Dear Authors,
Please, see below few comments you might consider or clarify:
Barriers to help-seeking are referenced to the wrong table; the cited table actually shows correlations. Ensure the barriers table is correctly numbered and titled, appears in sequence, and all in-text references point to the correct tables/figures.
Percentages for clinical symptoms and help-seeking use inconsistent ns across the text and tables. State the exact denominator for each percentage; report how many completed PHQ-9, GAD-7, and both; define the clinical thresholds used; and clarify whether “clinically significant” means either instrument above cut-off or both.
Results rely on many univariate tests without multiplicity control. Model key questions (especially predictors of not seeking help) using multivariable logistic regression (and ordinal models where appropriate), adjusting for relevant covariates. Report adjusted ORs with 95% CIs, and either control for multiple comparisons (e.g., Holm–Bonferroni) or pre-specify primary outcomes and mark the rest as exploratory.
Burnout is assessed with a single dichotomized item. Justify this choice with validation evidence and add sensitivity analyses using an alternative threshold. Clearly describe how coping behaviors were measured (response scale, frequency) and whether items were validated or author-derived.
Specify the exact item stems, response options, and whether barriers were checklist or free-text. If coding free-text, describe coder procedures and agreement. Present a table with counts and percentages for each barrier (multi-select clearly indicated).
Sample sizes vary across analyses. Provide an item-level missingness table, specify inclusion rules for each analysis (complete-case vs imputation), and consider multiple imputation if appropriate.
The sample is regional ED staff from an early pandemic window with a modest response rate. Reframe comparisons to other studies as descriptive; temper generalizations; and expand limitations on non-response, selection, timing, and local context.
Direction/sign of correlations for coping
Adaptive activities (family time, exercise) correlate positively with symptoms, which is counterintuitive. Clarify coding (higher = more time), discuss reverse causation/confounding, and consider partial correlations or adjusted regression including workload/role.
Add a post-hoc precision statement for key estimates and pre-specify a primary endpoint (e.g., help-seeking among those above symptom thresholds) to guide interpretation.
Grouping self-help reading with formal therapy/EAP obscures differences. Re-analyze with an ordered outcome (no action; informational only; group/peer; formal clinical care) and report predictors for formal care specifically.
Because supervisors disseminated reminders, describe anonymity safeguards and acknowledge potential social desirability or perceived pressure.
Avoid implying poor insight; note the alternative interpretation that self-assessment may be accurate when symptom scores are lower.
When comparing to external estimates, specify time windows, instruments, and whether populations are ED-specific vs mixed HCWs or general public.
Best wishes
Author Response
Please see attachment, thank you.

Reviewer 3 Report
Comments and Suggestions for Authors
Thank you very much for inviting me to review the manuscript titled “Frontline Healthcare Worker’s Reluctance to Access Psychological Support and Wellness Resources During COVID-19.” While the manuscript is well-structured, the issue is well-documented earlier.
- The title reflects the study type and essential components. Is it worker’s or workers’
- Abstract is fine. The authors need to mention explicitly statistical significance (p-values) when referring to key findings (e.g., comparisons between groups). The conclusion is too generic, and novelty is missing.
- Introduction: The study addresses a well-documented issue — the reluctance of healthcare workers (HCWs) to seek psychological support.
- Rationale and justification for the latest relevant issues are the least explored. Also, does not provide substantial new insights beyond what has already been reported extensively in the early COVID-19 literature (2020–2022).
- All the references are 2020 or earlier. This is not relevant to the current scenario.
- There is no exploration of longitudinal trends or changes in HCW attitudes over time, which would have strengthened novelty.
- Methods: A detailed method is missing. Authors must follow STROBE guidelines.
- No details about survey validation.
- Recruitment strategies are also poorly explained.
- It is unclear why the authors applied both parametric and non-parametric tests to a single dataset.
- Results: Low response rate (28.7%) is acknowledged but not analyzed for potential non-response bias.
- Why did the authors use numerical in some areas and numbers in others while writing the %?
- The discussion does not address how the results inform ongoing post-pandemic recovery, current HCW wellness strategies, or lessons for future crises.
- The conclusion is mainly descriptive and lacks forward thinking and novelty.
Author Response
Please see attached, thank you for your time

Round 2
Reviewer 2 Report
Comments and Suggestions for Authors
Thank you for addressing the comments
Author Response
Thank you for your work, helpful feedback, and efforts on the manuscript.
Reviewer 3 Report
Comments and Suggestions for Authors
No further comments.
Author Response

(The authors gave the same response as above.)
